# Which Specific Exercise Models Are Most Effective on Global Cognition in Patients with Cognitive Impairment? A Network Meta-Analysis

**DOI:** 10.3390/ijerph20042790

**Published:** 2023-02-04

**Authors:** Junchao Yang, Yunfeng Dong, Shuting Yan, Longyan Yi, Junqiang Qiu

**Affiliations:** 1School of Sport Science, Beijing Sport University, Beijing 100084, China; 2China Institute of Sports and Health, Beijing Sport University, Beijing 100084, China; 3Beijing Sports Nutrition Engineering Research Center, Beijing 100084, China

**Keywords:** exercise, cognitive impairment, global cognition, network meta-analysis

## Abstract

(1) Introduction: Physical exercise interventions can impart significant cognitive benefits to older adults suffering from cognitive impairment (CI). However, the efficacy of these interventions can vary widely, depending on the type, intensity, duration and frequency of exercise. (2) Aim: To systematically review the efficacy of exercise therapy on global cognition in patients with CI using a network meta-analysis (NMA). (3) Methods: The PubMed, Embase, Sport Discus (EBSCO) and Cochrane Library databases were electronically searched to collect randomized controlled trials (RCTs) on exercise for patients with CI from inception to 7 August 2022. Two reviewers independently screened the literature, extracted data, and assessed the risk of bias of the included studies. The NMA was performed using the consistency model. (4) Results: A total of 29 RCTs comprising 2458 CI patients were included. The effects of different types of exercise on patients with CI were ranked as follows: multicomponent exercise (SMD = 0.84, 95% CI 0.31 to 1.36, *p* = 0.002), short duration (≤45 min) (SMD = 0.83, 95% CI 0.18 to 1.19, *p* = 0.001), vigorous intensity (SMD = 0.77, 95% CI 0.18 to 1.36, *p* = 0.011) and high frequency (5–7 times/week) (SMD = 1.28, 95% CI 0.41 to 2.14, *p* = 0.004). (5) Conclusion: These results suggested that multicomponent, short-duration, high-intensity, and high-frequency exercise may be the most effective type of exercise in improving global cognition in CI patients. However, more RCTs based on direct comparison of the effects of different exercise interventions are needed. (6) NMA registration identifier: CRD42022354978.

## 1. Introduction

Mild cognitive impairment (MCI) and dementia are the two most common cognitive impairments (CI) [1,2]. Globally, there are approximately 50 million dementia patients, and the number is projected to reach 74.7 million by 2030 [3]. Dementia is one of the leading causes of disability and dependence among elderly individuals. It is estimated that the annual cost of caring for dementia patients will reach US $2 trillion by 2030. This undoubtedly places a heavy burden on society, especially since 60% of dementia patients originate in low- and middle-income countries [3,4]. MCI is a transitional state between normal cognitive function and dementia, and is thus often labelled as incipient dementia [5]; 14.9% of MCI patients (≥65 years of age) are likely to develop dementia within 2 years [6], while 16% of patients with amnestic mild cognitive impairment (aMCI) develop Alzheimer’s disease [1]. Only a minority of patients with MCI (14 to 40%) remain stable or return to normal [7]. Currently, scientific evidence supporting the use of pharmaceutical agents or supplements for symptomatic relief in dementia and mild cognitive impairment is highly insufficient [8,9].

Exercise has been shown to confer significant benefits on cardiovascular and cerebrovascular function associated with dementia, but evidence of improved cognitive function is limited, especially in cognitively normal older adults [10]. Nonetheless, an increasing number of studies have proven that exercise can enhance cognitive function in people who suffer from cognitive impairment. Studies comparing the efficacy of exercise therapy in patients with cognitive impairment have focused on investigating and comparing the efficacy of the various types of physical activity. A meta-analysis by Panza and his colleagues [11], for example, postulated that aerobic exercise is more beneficial for Alzheimer’s patients. According to Northey et al. [12], tai chi is more effective for elderly individuals (over 50 years old) with dementia or severe mental illness. In another network meta-analysis (NMA), resistance exercise was found to be the most effective method for improving overall cognitive function in MCI patients [13]. Nevertheless, there is inadequate research on the relationships between intensity, duration, or frequency of exercise and cognitive boost. This study aims to conduct a network meta-analysis on the various exercise regimens, and thus contribute to filling that gap.

A network meta-analysis can compare not only the types of exercise that are more effective, but also the intensity, time, and frequency at which it is most effective. Therefore, randomized controlled trials (RCTs) of exercise intervention in patients with CI were the focus in this meta-analysis. It is our hope that this work will evaluate the effectiveness of exercise prescriptions in the literature, consequently providing scientific evidence to guide the prescription of exercise for patients with cognitive impairment.

## 2. Materials and Methods

The NMA was conducted according to Preferred Reporting Items for Systematic Reviews and Meta-Analyses (PRISMA) guidelines [14].

### 2.1. Eligibility Criteria and Study Selection

This NMA included studies that met the following criteria: (i) Population: adults were over 50 years old. Since the original authors had differing criteria for diagnosing cognitive ability from study to study, the participants were diagnosed as having CI (e.g., MCI, dementia) as interpreted by the original authors [12]. (ii) Intervention: relevant interventions included either exercise alone or in combination with a control group. The specific exercises were classified as follows: (a) type: aerobic, resistance training, multicomponent training, or mind-body exercise; (b) intensity: light, moderate, or vigorous (c) time: short (≤45 min), medium (>45 to ≤60 min), or long (>60 min); and (d) frequency (number of exercise sessions per week): low (≤2), medium (3–4), or high (5–7). (iii) Control: comparators for non-exercise training treatments included true control, health education, or stretching. (iv) Outcome: global cognition measured by any validated neuropsychological test. (v) Type of study: randomized controlled trials. The following are the exclusion criteria: prospective or retrospective cohort studies, case reports, conference abstracts, and literature not written in English.

### 2.2. Search Strategy

A search was performed in PubMed, Embase, Sport Discus (EBSCO), and The Cochrane Library for work published before August 2022. The following keywords were chosen to screen studies: cognitive dysfunction, mild cognitive impairment, dementia, Alzheimer’s disease, mild cognitive disorder, aerobic, walking, high intensity interval, resistance, core stability, dance, breathing exercise, virtual reality exercise, whole-body vibration exercise, stretching, body-mind exercise, yoga, pilates, tai chi, taijiquan, health qigong, yijinjing, wuqinxi, liuzijue, baduanjin, and multicomponent exercise. The PubMed search strategy is presented in Table 1. The bibliographies of the selected studies and reviews were painstakingly reviewed to ensure that relevant literature was not missed.

### 2.3. Data Extraction

Two reviewers independently screened the abstracts and full-text articles of these selected works, extracted and cross-checked the data. In case of a disagreement, a third party was consulted to mediate it, and a consensus was reached. In the event that an article lacked key data, the corresponding author was contacted for clarification. During literature screening, the title and abstract were read first, and then the full text was read, to identify literature that would be eliminated and that which would be kept. The following data were extracted from the selected works: research title; name of corresponding author; publication journal and time; gender of study participants; type of cognitive impairment; sample size; intervention used; control measures; type, intensity, frequency, duration, and time of exercise therapy; follow-up time after intervention; tools for measurement of cognitive function; relevant patient outcomes; and risk of literature bias.

### 2.4. Risk of Bias of Individual Studies

Two reviewers assessed the risk of bias in the included studies according to the Cochrane Risk of Bias Tool (ROB-2) for RCTs. Funnel plots were also generated, and any asymmetry examined. As this study involved exercise interventions, it was not possible to blind patients to treatment allocation. Blinding of participants and personnel was thus deemed to be a high risk of bias in all studies and was not factored into the overall risk of bias assessment. The evaluation included six items on random sequence generation, allocation concealment, blinding of outcome assessment, incomplete outcome data, selective reporting, and other bias. Each item was rated as a “high risk”, “unclear risk”, or “low risk” of bias.

### 2.5. Data Analysis

STATA15.1 (StataCorp, College Station, TX, USA) was used to perform the statistical analysis within a frequentist framework. Standardized mean difference (SMD) was used as the summary statistic in studies that objectively measured global cognition outcomes using a variety of unit types. In this meta-analysis, the I2 statistic was used to rate heterogeneity into three categories: low (less than 25%), moderate (25% to 50%), or high (more than 50%). Consistency—meaning that the treatment effects estimated from direct comparisons agree with those estimated from indirect comparisons—was assessed by fitting both a consistency and an inconsistency NMA and examining the results from the Wald-type chi-squared test for inconsistency. Once the effectiveness of the interventions was compared, the interventions were ranked using the surface under the cumulative ranking curve (SUCRA). High odds in SUCRA provide better results for an intervention [15]. Thus, in a ranking table, we reported the pairwise effect size (ES) comparisons above the leading diagonal, and the ES estimated from NMA [16] below the leading diagonal.

## 3. Results

### 3.1. Literature Selection

A total of 6631 RCTs were initially identified. Following a review of the titles and abstracts, 89 studies were selected for full manuscript review. Of these, 60 were excluded because they did not fulfil our inclusion criteria. The remaining 29 studies [17,18,19,20,21,22,23,24,25,26,27,28,29,30,31,32,33,34,35,36,37,38,39,40,41,42,43,44,45] were included in this review. The PRISMA chart is illustrated in Figure 1.

### 3.2. Characteristics of the Included RCTs

The characteristics of the included studies are presented in Table 2. The sample size ranged from 19 to 415 patients (mean age range, 55–85 years; two studies did not report age), and the study duration ranged from 8 to 48 weeks. Exercise training interventions included multicomponent exercise (studies: *n* = 9, patients: *n* = 483), resistance training (studies: *n* = 5, patients: *n* = 87), mind-body exercise (studies: *n* = 3, patients: *n* = 143), and aerobic exercise (studies: *n* = 14, patients: *n* = 559). The frequency of exercise ranged from 1 to 5 days each week, while each session of physical activity lasted for 15 to 90 min. The Mini-Mental State Examination was the most widely used cognitive measurement tool. Other tools noted were the Montreal Cognitive Assessment (MoCA), Alzheimer’s Disease Assessment Scale (ADAS-Cog), and Neurobehavioral Cognitive Status Examination (NCSE).

The risk of bias assessment for each individual study is presented in Figure 2. Overall, the studies tended to exhibit a low risk of random sequence generation (62%), allocation concealment (52%), blinding of outcome assessment (56%), incomplete outcome data (100%), and selective outcome reporting (48%), but not blinding of patients and personnel (0%) or other bias (0%).

### 3.3. Network Meta-Analysis

Four types of exercise therapy were included in this study, namely, multicomponent exercise, resistance training, mind-body exercise, and aerobic exercise. In addition to the direct comparison between multicomponent exercise and resistance training, there were direct comparisons between other exercise types, the most direct studies being conducted between the aerobic exercise and control groups (Figure 3a). Low-intensity exercise, medium-intensity exercise, and high-intensity exercise were also directly contrasted. There was no direct comparison between short, medium, and long periods of exercise (Figure 3b), with the most direct studies being between the short-duration and control periods (Figure 3c). There was no direct comparison between low-, medium-, and high-frequency exercise, with the most direct studies being between moderate and control exercise (Figure 3d). All pairwise comparisons in an NMA are included in the comparison league tables (Table 3).

#### 3.3.1. Exercise Type

The results from the consistency NMA provided evidence that, when compared with the control, multicomponent exercise (*p* = 0.002), aerobic exercise (*p* = 0.001), and resistance training (*p* = 0.037) all resulted in greater improvements in global cognition following intervention. In contrast, the results from the consistency NMA did not provide evidence for a significant impact of mind-body exercise on global cognitive function. The results of the consistency model indicated that multicomponent exercise (SUCRA = 74.3%), aerobic (SUCRA = 63.9%) and resistance training (SUCRA = 61.0%) were among the best interventions for global cognition (Table 4).

#### 3.3.2. Exercise Time

The results of the consistency NMA indicated that, as with the control, short (*p* < 0.001) and medium (*p* = 0.008) exercise duration resulted in enhanced global cognition following intervention. In contrast, the results from the consistency NMA did not provide evidence for a significant impact of prolonged exercise on global cognition. The results of the consistency model indicated that exercise of short (SUCRA = 77.3%) or medium (SUCRA = 62.9%) duration yielded the best results (Table 4).

#### 3.3.3. Exercise Intensity

The results of the consistency NMA indicated that, as with the control, vigorous (*p* < 0.011) and moderate (*p* = 0.001) exercise intensity resulted in enhanced global cognition following intervention. In contrast, the results from the consistency NMA did not provide evidence for a significant impact of light exercise on global cognition. The results of the consistency model indicated that exercise of vigorous (SUCRA = 74.4%) or moderate (SUCRA = 71.8%) intensity yielded the best results (Table 4).

#### 3.3.4. Exercise Frequency

The results from the consistency NMA provided evidence that, when compared with the control, high (*p* = 0.004), medium (*p* = 0.001), and moderate (*p* = 0.011) exercise frequencies all resulted in enhanced global cognition.

The results of the consistency model indicated that vigorous (SUCRA = 91.3%), medium (SUCRA = 64.0%), and moderate (SUCRA = 44.5%) exercise frequencies were among the best interventions for global cognition (Table 4).

### 3.4. Publication Bias

According to the funnel plot of this study, there was either publication bias or a small sample effect.

## 4. Discussion

This NMA confirmed that exercise can preserve global cognitive function in patients with cognitive impairment. The optimal type, intensity, duration, and frequency of exercise have also been established. It must be noted that the quality of the RCTs included in this analysis was relatively high. The following were other key merits of this analysis: a broad timeline of studies, highly international authorship, and exclusion of biased RCTs. Therefore, the results of this NMA can be considered relatively robust and of high clinical value.

According to this study, multicomponent, aerobic, resistance, and mind-body exercises all improve cognition in patients with CI (in descending order of efficacy). A subgroup analysis performed by Northey et al. [12] found that tai chi exercise is the most effective, followed by multimodal resistance exercise and aerobic exercise, but deemed hatha yoga as ineffective. The results of our study conflict with those from the aforementioned study, likely because we categorized yoga and tai chi as mind-body exercises [46] and excluded the cognitively “normal” population from our study. According to Northey and his colleagues [12], resistance exercise may be the most effective form of physical activity for MCI patients. In this study, aerobic dance was classified as mind-body exercise, and multimodal exercise was not included, which may explain the inconsistency with our study.

In our study, exercise of short duration (≤45 min) was 34.54 ± 8.65 min. It is interesting to note that the study conducted by Cancel et al. [26] had the shortest exercise (15 min), used a powerful bicycle, and suffered little resistance during their exercise. It was found in our study that vigorous intensity exercise was most effective, while the analysis by Northey et al. [12] showed that moderate and vigorous intensity exercise had similar success. This contrast can be primarily attributed to the fact that our meta-analysis involved only patients with cognitive impairments. The results of our study are corroborated by previous studies that have shown highfrequency exercise to be the most effective [12]. The conclusions on optimal exercise duration, frequency, and intensity made in this study are in line with the recommended ACSM physical activity guidelines [47]: 75–150 min of high-intensity aerobic exercise per week, with additional resistance exercise (≥2 times/week).

In spite of the many qualities of this NMA, a few limitations must be acknowledged: (1) Although the subjects and researchers in the reviewed RCTs could not be blinded, most of these studies blinded the outcome measures; this may result in a selective bias; (2) There is a large difference in the number of studies under each moderating variable; in particular, there is no direct comparative study on exercise intensity and frequency, manifesting inconsistencies in the results of future direct comparison studies; (3) The synergistic effects of exercise type, time, intensity, and frequency on cognitive function were not considered; (4) Publication bias in the meta-analysis (as only RCTs in English were used); (5) The number of RCTs included is arguably low (*n* = 29). The meta-analysis thus has low statistical power.

## 5. Conclusions

Exercise of a multicomponent, short-duration, high-intensity, and high-frequency nature may yield the best cognitive benefits for patients with cognitive impairment. In the future, more RCTs can be conducted based on direct comparison of the effects of different exercise interventions.

## Figures and Tables

**Figure 1 ijerph-20-02790-f001:**
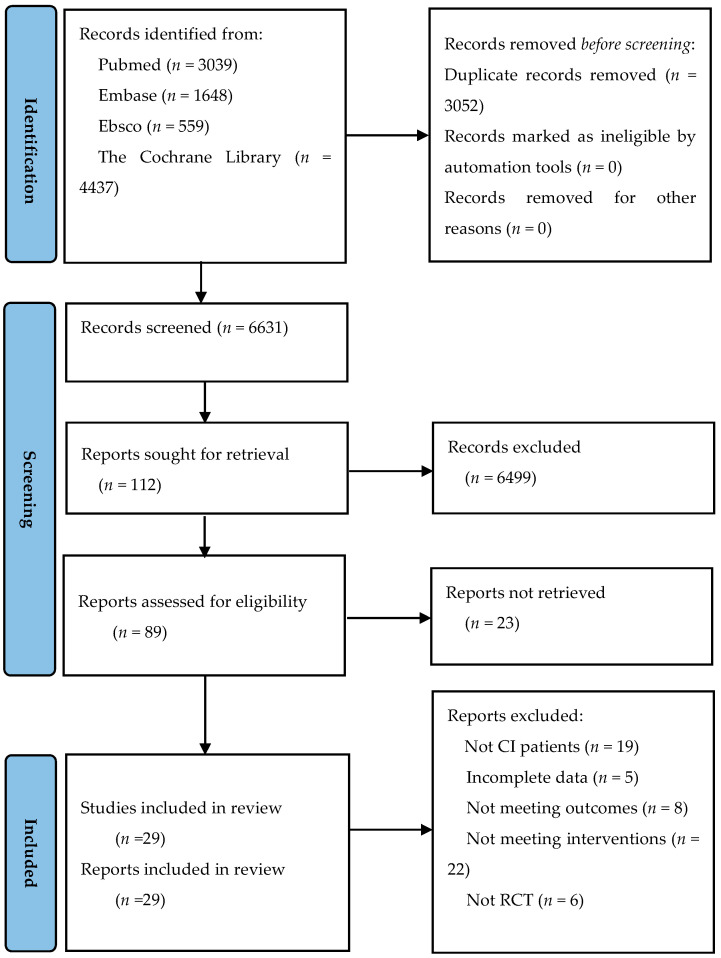
PRISMA flow diagram of included studies.

**Figure 2 ijerph-20-02790-f002:**
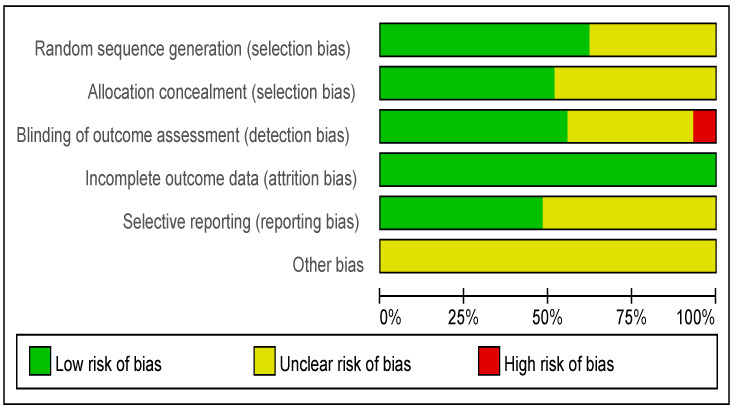
Risk of bias graph.

**Figure 3 ijerph-20-02790-f003:**
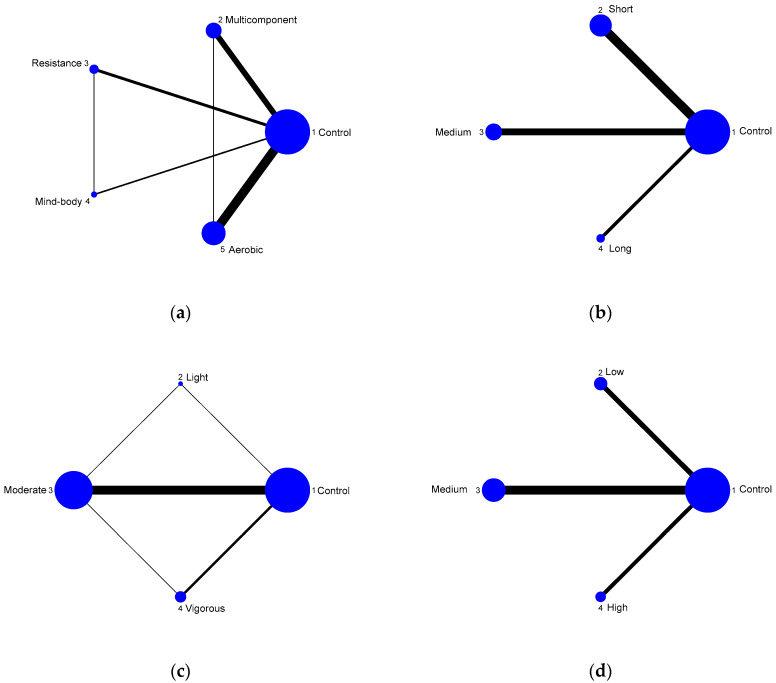
NMA maps of the studies examining the efficacy of exercise in people living with CI on global cognition. (**a**) exercise type; (**b**) exercise time; (**c**) exercise intensity; (**d**) exercise frequency.

**Table 1 ijerph-20-02790-t001:** PubMed search strategy.

Serial Number	Content
#1	Cognitive Dysfunction [Mesh Terms] OR Dementia [Mesh Terms] OR Alzheimer Disease [Mesh Terms]
#2	Mild Cognitive Impairment [Title/Abstract] OR Mild Cognitive Disorder [Title/Abstract] OR Mild Cognitive Dysfunction [Title/Abstract] OR Mild Cognitive Decline [Title/Abstract] OR Mild Neurocognitive Disorder [Title/Abstract]
#3	#1 OR #2
#4	Sports [Mesh Terms] OR Exercise [Mesh Terms] OR Exercise Movement Techniques [Mesh Terms] OR Resistance Training [Mesh Terms] OR Yoga [Mesh Terms] OR Dance Therapy [Mesh Terms] OR Virtual Reality Exposure Therapy [Mesh Terms] OR Breathing Exercises [Mesh Terms] OR Walking [Mesh Terms] OR High-Intensity Interval Training [Mesh Terms] OR Muscle Stretching Exercises [Mesh Terms] OR Tai Ji [Mesh Terms]
#5	Core Stability [Title/Abstract] OR Whole Body Vibration Exercise [Title/Abstract] OR Body-Mind Exercise [Title/Abstract] OR Health Qigong [Title/Abstract] OR Yijinjing [Title/Abstract] OR Wuqinxi [Title/Abstract] OR Liuzijue [Title/Abstract] OR Baduanjin [Title/Abstract] OR Multicomponent Exercise [Title/Abstract]
#6	#4 OR #5
#7	#3 AND #6

**Table 2 ijerph-20-02790-t002:** Summary of studies included in the NMA.

Study	Age (T/C)	Sample (T/C)	Intervention/Comparator	Exercise Moderators				Outcome
				Intensity	Time (Min)	Frequency (Days/Week)	Duration (Weeks)	
Lam 2012 [17]	77.20 ± 6.30/78.30 ± 6.60	93/169	Tai chi/stretching	moderate	30	≥3	48	①③
Suzuki 2013 [18]	74.80 ± 7.40/75.80 ± 6.10	47/45	multicomponent exercise/education	moderate	90	2	24	①③
Yu 2021 [19]	77.4 ± 6.6/77.5 ± 7.1	64/32	cycling/stretching	moderate	40–60	3	24	①
Bisbe 2019 [20]	72.88 ± 5.60/77.29 ± 5.16	17/14	aerobic dances/multicomponent exercise	Moderate/moderate	60	NR	12	③
Lamb 2018 [21]	≥65	278/137	aerobic dances/usual care	NR	60–90		12	①
Yoon 2016 [22]	NR	23/7	strength training/stretching	vigorous/moderate	60	2	12	③
Langoni 2018 [23]	72.60 ± 7.80/71.90 ± 7.90	26/26	multicomponent exercise/no exercise	moderate	60	2	24	③
Wei 2014 [24]	66.73 ± 5.48/65.27 ± 4.63	30/30	handball/recreational activities	moderate	30	5	24	③
Hong 2017 [25]	75.92 ± 4.81/77.89 ± 3.40	10/12	resistance exercise/maintain current lifestyle	moderate	60	2	12	④
Cancela 2016 [26]	80.63 ± 8.32/82.90 ± 7.42	73/113	aerobic exercise/recreational activities	moderate	15	4	12	③
Song 2019 [27]	76.22 ± 5.76/75.33 ± 6.78	60/60	aerobic exercise/education	moderate	60	2	16	③
Huang 2019 [28]	81.90 ± 6.00/81.90 ± 6.10	36/38	Tai Chi/usual care	moderate	20	3	40	④
Kemoun 2010 [29]	81.80 ± 5.30	16/15	multicomponent exercise/recreational activities	moderate	60	3	15	②
Jurakic 2017 [30]	70.40 ± 3.93	14/14	resistance exercise/Pilates	moderate/moderate	30–60	3	8	④
Law 2019 [31]	77.94 ± 6.11/75.14 ± 8.53	16/14	multicomponent exercise/maintain current lifestyle	moderate	60	2	8	⑤
Shimada 2018 [32]	70.10 ± 4.00/70.70 ± 4.70	53/47	golf/education	moderate	90–120	1	12	③
Bademli 2019 [33]	72.24 ± 7.16/70.67 ± 8.34	30/30	multicomponent exercise/maintain current lifestyle	moderate	80	3	20	③
Holthoff 2015 [34]	72.40 ± 4.34/70.67 ± 5.41	13/12	resistance exercise/maintain current lifestyle	moderate	30	3	24	③
Varela 2011 [35]	77.88 ± 10.75/79.40 ± 6.72	33/15	cycling/recreational activities	low	30	3	12	③
Lazarou 2017 [36]	67.92 ± 9.47/65.89 ± 10.76	66/63	international ballroom dancing/no exercise	moderate	60	2	40	③④
Mavros 2017 [37]	NR	27/23	resistance exercise/stretching	moderate	75	2	24	①
Hoffmann 2016 [38]	69.80 ± 7.40/71.30 ± 7.30	102/88	aerobic exercise/maintain current lifestyle	vigorous	60	3	16	③
Li 2021 [39]	≥60	42/42	multicomponent exercise/stretching	NR	30	5	24	③④
Tomoto 2021 [40]	55–80	17/20	multicomponent exercise/stretching	vigorous	30–40	3–4	48	③
Aguiar 2014 [41]	74.70 ± 7.40/78.60 ± 8.40	17/17	multicomponent exercise/usual care	moderate	40	2	24	③
Venturelli 2011 [42]	83 ± 6/85 ± 5	11/10	walking program/usual care	moderate	30	3	24	③
Yang 2015 [43]	72.00 ± 6.69/71.92 ± 7.28	25/25	aerobic exercise/education	vigorous	40	3	12	①
Silva 2019 [44]	71.85 ± 5.69/78.20 ± 5.26	12/7	multicomponent exercise/maintain current lifestyle	moderate	60	2	12	③
Arcoverde 2014 [45]	78.5 (64–81.2)/79 (74.7–82.2)	10/10	aerobic exercise/education	vigorous	30	2	16	③

NR: not reported; T/C: intervention/comparator; ① ADAS-Cog; ② ERFC; ③ MMSE; ④ MoCA; ⑤ NCSE.

**Table 3 ijerph-20-02790-t003:** Ranking table of exercise moderators.

Moderators	Effect Size				
exercise type	multicomponent exercise	−0.05 (−1.68, 1.59)	-	-	−0.85 (−1.41, −0.28)
	0.11 (−0.53, 0.75)	aerobic exercise	0.82 (−0.84, 2.48)	-	−0.72 (−1.55, 0.12)
	0.03 (−0.89, 0.96)	−0.08 (−0.94, 0.79)	resistance training	-	−0.48 (−1.55, 0.59)
	0.50 (−0.55, 1.55)	0.39 (−0.61, 1.40)	0.47 (−0.57, 1.50)	mind-body exercise	−0.72 (−1.17, −0.28)
	0.84 (0.31, 1.36)	0.73 (0.31, 1.15)	0.80 (0.05, 1.56)	0.34 (−0.58, 1.25)	control
exercise time (min)	short	-	-	-	
	0.03 (−0.42, 0.47)	medium	-	-	
	0.27 (−0.29, 0.83)	0.24 (−0.33, 0.82)	long	-	
	0.83 (0.53, 1.13)	0.80 (0.48, 1.13)	0.56 (0.09, 1.04)	control	
exercise intensity	vigorous	-	-	-	
	0.04 (−0.60, 0.69)	moderate	-	-	
	0.35 (−0.96, 1.65)	0.30 (−0.87, 1.47)	light	-	
	0.77 (0.18, 1.36)	0.72 (0.43, 1.02)	0.42 (−0.75, 1.59)	control	
exercise frequency (days/week)	high	-	-	-	
	0.47 (−0.51, 1.46)	medium	-	-	
	0.69 (−0.29, 1.67)	0.22 (−0.44, 0.87)	low	-	
	1.28 (0.41, 2.14)	0.80 (0.33, 1.27)	0.59 (0.14, 1.04)	control	

**Table 4 ijerph-20-02790-t004:** Results of the NMA and ranking.

Moderators		Sample Size (Number of Studies)	SMD (95% CI)	*p* Value	Rank (SUCRA)
exercise type	multicomponent exercise	483 (9)	0.84 (0.31, 1.36)	0.002	1 (74.3)
	resistance training	87 (5)	0.80 (0.05, 1.56)	0.037	3 (61.0)
	mind-body exercise	143 (3)	0.34 (−0.58, 1.25)	0.471	4 (44.4)
	aerobic exercise	559 (14)	0.73 (0.31, 1.15)	0.001	2 (63.9)
exercise time (min)	short (≤45)	400 (12)	0.83 (0.18, 1.19)	<0.001	1 (77.3)
	medium (>45–≤60)	378 (10)	0.68 (0.48, 1.13)	0.008	2 (62.9)
	long (>60)	435 (5)	0.65 (−0.04, 1.34)	0.067	3 (58.5)
exercise intensity	light	17 (1)	0.42 (−0.75, 1.59)	0.478	3 (45.9)
	moderate	681 (20)	0.72 (0.43, 1.02)	<0.001	2 (71.8)
	vigorous	93 (5)	0.77 (0.18, 1.36)	0.011	1 (74.4)
exercise frequency (days/week)	low (≤2)	629 (12)	0.59 (0.14, 1.04)	0.011	3 (44.5)
	medium (3–4)	437 (12)	0.80 (0.33, 1.27)	0.001	2 (64.0)
	high (5–7)	145 (3)	1.28 (0.41, 2.14)	0.004	1 (91.3)

## Data Availability

Further inquiries should be directed to the corresponding author.

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
