# Peer review of "Which Specific Exercise Models Are Most Effective on Global Cognition in Patients with Cognitive Impairment? A Network Meta-Analysis"

_ijerph, 2023, doi:10.3390/ijerph20042790_

Round 1

Reviewer 1 Report

1. The study focus on "cognitive impairment", but author wrote about "Mild cognitive impairment" . There is a significant gap between these two words. Please consider the word used in revised version.

2. The current study, author analyzed the data from RCTs' study. But there were no specific word on it in the "Search strategy". Please provide reason.

3. Did all the "cognitive impairment" was dignosed by doctor?or by some screen questionnaire? Which is very important for current study. Please provide more detail information.

Author Response

Reviewer #1:

Point 1: The study focus on "cognitive impairment", but author wrote about "Mild cognitive impairment". There is a significant gap between these two words. Please consider the word used in revised version.

Response 1: It is now revised (line 11-13), thank you.

Point 2: The current study, author analyzed the data from RCTs' study. But there were no specific word on it in the "Search strategy". Please provide reason.

Response 2: As much as possible, we didn't restrict RCTs in the development of search strategy in order to collect literature related to our research purpose. 

Point 3:  Did all the "cognitive impairment" was dignosed by doctor?or by some screen questionnaire?Which is very important for current study. Please provide more detail information.

Response 3: Thank you for pointing this out. We have added the details of diagnosing CI in red (line 150).

Reviewer 2 Report

Dear authors,

Thank you for the opportunity to review your manuscript. It is a work that focuses on a topic of intriguing importance, especially in light of the power of statistical intervention:

Abstract:

I would suggest being blunter in defining cognitive impairment… mild or severe? Or remove and better describe population eligibility in methods.

Since you mention the use of Risk of Bias, which one? RoB-2? Joanna Briggs Inst.?

18 it is not necessary to mention the software in the abstract

23 these results suggested that..

31, however, are incomparable... in fact then in the PICO, you speak of "actual" dementia not of mild cognitive impairment, please correct throughout the manuscript

Remove the box, make a table

117 RoB-2?

In statistical analysis and manuscript I would suggest adding the net league tables and references “Thus in a ranking table, we will report above the leading diagonal the pairwise effect size comparisons, below the leading diagonal ESs estimated from network meta-analyses” Reference to:   https://pubmed.ncbi.nlm.nih.gov/32478581/

In addition I recommend a description of SUCRA's ability to provide the best possible option out of a range of possibilities, for example: “High odds in SUCRA will provide better results for an intervention”

Ref: https://doi.org/10.1016/j.rehab.2021.101602

The results are not well-structured, figure 1 should be placed first because it conveys the selection process, then certainly the table with the characteristics of the studies.

Then all netplots with the description of line thicknesses and circle diameters, net leagues with pairwise and network comparisons. Finally, the rankings through the sucra.

In fact, there are no errors but the manuscript is difficult to approach as formulated

Author Response

Reviewer #2:

Thank you for the opportunity to review your manuscript. It is a work that focuses on a topic of intriguing importance, especially in light of the power of statistical intervention:

Point 1: Abstract: I would suggest being blunter in defining cognitive impairment… mild or severe? Or remove and better describe population eligibility in methods.

Response 1: Reply: Thank you very much.We have added the details of diagnosing CI in red (line 150).

Point 2: Since you mention the use of Risk of Bias, which one? RoB-2? Joanna Briggs Inst.?

Response 2: We assessed the risk of bias in the included studies according to the Cochrane Risk of Bias Tool (ROB-2) for RCTs (line 118).

Point 3: 18 it is not necessary to mention the software in the abstract

Response 3: Revised as suggested (line 18).

Point 4: 23 these results suggested that..

Response 4: Revised as suggested (line 23).

Point 5: 31, however, are incomparable... in fact then in the PICO, you speak of "actual" dementia not of mild cognitive impairment, please correct throughout the manuscript

Response 5: Thank you for the suggestion. We have now revised as suggested. In fact, we wanted to include all patients with cognitive impairment, regardless of the level, and we restated the inclusion criteria.

Point 6: Remove the box, make a table

Response 6: Correction has been made.

Point 7: 117 RoB-2?

Response 7: Yes, revised as suggested. 

Point 8: In statistical analysis and manuscript I would suggest adding the net league tables and references “Thus in a ranking table, we will report above the leading diagonal the pairwise effect size comparisons, below the leading diagonal ESs estimated from network meta-analyses” Reference to:   https://pubmed.ncbi.nlm.nih.gov/32478581/

Response 8: Thank you for the suggestion. We have added a ranking table In statistical analysis and manuscript (line 143 and line 207).

Point 9: In addition I recommend a description of SUCRA's ability to provide the best possible option out of a range of possibilities, for example: “High odds in SUCRA will provide better results for an intervention”

Ref: https://doi.org/10.1016/j.rehab.2021.101602

Response 9: We have added the a description of SUCRA's ability in methods (line 142).

Point 10: The results are not well-structured, figure 1 should be placed first because it conveys the selection process, then certainly the table with the characteristics of the studies.

Then all netplots with the description of line thicknesses and circle diameters, net leagues with pairwise and network comparisons. Finally, the rankings through the sucra.

In fact, there are no errors but the manuscript is difficult to approach as formulated

Response 10: Thank you for pointing this out. We have placed figure 1 first and then all netplots with the description. However, we plan “net leagues with pairwise and network comparisons” and “rankings through the sucra” together finally.

Ref: http://dx.doi.org/10.1136/bjsports-2019-100886.

Round 2

Reviewer 1 Report

I do not have further more comments.

Reviewer 2 Report

Dear Authors,

I believe that the manuscript has reached an appreciable methodological level and therefore I suggest its suitability for publication